# Machine Learning Modeling from Omics Data as Prospective Tool for Improvement of Inflammatory Bowel Disease Diagnosis and Clinical Classifications

**DOI:** 10.3390/genes12091438

**Published:** 2021-09-18

**Authors:** Biljana Stankovic, Nikola Kotur, Gordana Nikcevic, Vladimir Gasic, Branka Zukic, Sonja Pavlovic

**Affiliations:** Laboratory for Molecular Biomedicine, Institute of Molecular Genetics and Genetic Engineering, University of Belgrade, 11042 Belgrade, Serbia; nikola.kotur@imgge.bg.ac.rs (N.K.); gordnik@imgge.bg.ac.rs (G.N.); vlada.gasic@imgge.bg.ac.rs (V.G.); branka.zukic@imgge.bg.ac.rs (B.Z.); sonya@sezampro.rs (S.P.)

**Keywords:** IBD, artificial intelligence, prediction modeling, genomics, transcriptomics

## Abstract

Research of inflammatory bowel disease (IBD) has identified numerous molecular players involved in the disease development. Even so, the understanding of IBD is incomplete, while disease treatment is still far from the precision medicine. Reliable diagnostic and prognostic biomarkers in IBD are limited which may reduce efficient therapeutic outcomes. High-throughput technologies and artificial intelligence emerged as powerful tools in search of unrevealed molecular patterns that could give important insights into IBD pathogenesis and help to address unmet clinical needs. Machine learning, a subtype of artificial intelligence, uses complex mathematical algorithms to learn from existing data in order to predict future outcomes. The scientific community has been increasingly employing machine learning for the prediction of IBD outcomes from comprehensive patient data-clinical records, genomic, transcriptomic, proteomic, metagenomic, and other IBD relevant omics data. This review aims to present fundamental principles behind machine learning modeling and its current application in IBD research with the focus on studies that explored genomic and transcriptomic data. We described different strategies used for dealing with omics data and outlined the best-performing methods. Before being translated into clinical settings, the developed machine learning models should be tested in independent prospective studies as well as randomized controlled trials.

## 1. Introduction

Inflammatory bowel disease (IBD) is a complex disease, characterized as chronic, relapsing and remitting intestinal inflammation, with substantial heterogeneity among clinical phenotypes with regards to the age at diagnosis, severity of symptoms, response to therapy and long-term clinical outcomes [1,2,3]. It has traditionally been considered to comprise two major subtypes, Crohn’s disease (CD) and ulcerative colitis (UC) [4]. This classification is mainly based on distinctive parameters primarily related to the location and behavior of the disease. CD can occur at various parts of the gastrointestinal (GI) tract-from mouth to anus, and it is patchy, transmural, and may have inflammatory (also called nonstricturing/nonpenetrating), stricturing or penetrating (fistulating) behavior [5]. UC is typically restricted to the colon and rectal mucosa, without fibrotic strictures [6,7]. In addition to CD and UC, there are patients whose disease characteristics cannot fit precisely into either of these two subtypes, which are described as ‘IBD unclassified’ (IBDU), and they are more common in children [8].

A number of interacting factors are accountable for the pathogenesis of IBD, of which genetic susceptibility, bacterial recognition and immune response of the host, microbiota and diet are among the most significant ones [9,10].

The search for IBD genetic determinants resulted in identification of more than 240 gene loci that have been associated with an increased risk of developing this disease [11,12,13]. In 2001 the frameshift variant of *NOD2* (nucleotide-binding oligomerization domain-containing protein 2) gene was identified as the first CD susceptibility genetic variant [14]. Currently, IBD is characterized as a polygenic disease, driven by multiple common genetic variants, of which *NOD2* variants have the highest effect size [12,15]. Also, it has been shown that rare monogenic variants contribute to the IBD risk, and to date, around 50 single genes are implicated in very-early-onset IBD [12,15,16].

Further studies revealed that the major perturbed molecular processes in IBD are associated with signaling pathways involved in innate and adaptive immune response, autophagy and intestinal epithelial barrier function and repair [17,18,19,20,21]. Although the etiology of IBD still remains undefined, the host-genome association with gut microbiome is in the focus of the current model of IBD pathogenesis. It is based on the concept of misdirected response of the host’s immune system to intestinal microbial and immunogenic factors that involve the inflammation-associated mucosal injury [22]. It is believed that these are the key steps which promote disease severity, relapse and also its progression to neoplastic transformation [12,23,24].

Currently, there is no cure for IBD, and in a significant number of cases, applied therapies are found to be ineffective or lead to a poor/inadequate response [25,26,27]. In addition, it is often not possible to establish an accurate diagnosis of IBD since it depends on a combination of numerous clinical data, including complex image assessments, whose interpretation is inherently subjective [28]. Altogether, untimely and inaccurate diagnosis has a great impact on the course of the disease, which usually leads to complications and thus represents a serious obstacle to achieving and maintaining remission of the disease, which is the main goal of the IBD treatment [29]. Diagnosis, classification, prognosis and therapy of IBD still require the detection of accurate and reliable biomarkers and their translation into clinical practice, with the aim to significantly improve outcomes in patients with IBD.

Regarding molecular classification of disease subtypes, it has been shown that most of detected IBD loci confer risk to both CD and UC, typically with distinct effect sizes in each disorder; whereas the minor number of loci is unique to each subtype [15,30]. In addition to the latter, there are examples, such as variants at the *NOD2* and protein tyrosine phosphatase nonreceptor type 22 (*PTPN22*) loci, which have been found to be risk factors for CD, while for UC they have been shown to be variants with a protective effect [30]. These results provided evidence for fundamental etiological differences between the two IBD subtypes [21]. In addition, important connections have been found between molecular and clinical phenotypes, such as the ones related to disease location (associations of *NOD2*, *MHC* and *MST1* 3p21 variants with ileal vs. colonic CD) and to disease behavior (associations of *NOD2*, *MHC* and *MST1* 3p21 variants with deep ileal ulcers; *NOD2* variants with fibrostenotic or stricturing ileal CD; miR-215 expression with penetrating CD and differential DNA methylation with inflammation in CD) [21].

Cumulative evidence suggests that classifying IBD as CD or UC might be oversimplified and that particular disease phenotypes could also be considered as genetically distinct entities [11]. Namely, results of the large genotype-subphenotype study performed on the Immunochip array showed that colonic CD was genetically intermediate between ileal CD and UC; i.e., predictive models based on genetic risk scores identified that ileal CD and colonic CD are as different from each other as they are from UC [11]. This finding corroborated the results of several earlier studies, which have found that *NOD2* gene variants are associated with ileal Crohn’s disease, thus delineating it from colonic CD, with a shorter time for the onset of stenosing disease and need for surgery [12]. Important additional evidence is that differential microRNA (miRNA) expression was detected between these entities [31]. In this sense, it has been suggested recently that ileal and colonic CD could potentially be regarded as separate diseases and that consideration should be given to a new classification for CD, which splits it into ileum dominant (isolated ileal and ileocolonic) and isolated colonic disease. This may allow for a more optimized approach to clinical care and scientific research for CD [32].

The integration of data from expression (mRNA, miRNA and protein) and epigenetic (DNA methylation and histone modifications) examinations is progressively more present in IBD studies, which significantly contribute to the improvement of IBD classification, reduce misdiagnoses and assist clinical decisions regarding the choice of adequate therapies [21,33,34,35,36]. The emergence of new high-throughput technologies enabled the usage of genomic, transcriptomic, epigenomic, proteomic, metabolomic, metagenomic, in general—omics data, for the purpose of achieving the goals of precision medicine. The clinical diagnostics based on omics analysis using next generation sequencing are applicable, especially in the fields of inherited disease diagnosis and oncology. Several omics-based tests have been FDA (Food and Drug Administration) approved for the clinical application in the USA, as well as certified with CE (Conformitè Europëenne) marking for clinicians in Europe. Most of the tests comprise genome analysis, but there are also tests using RNA whole-transcriptome sequencing (https://www.clinicalomics.com, accessed on 25 November 2020).

The analysis of omics big data demands the usage of powerful bioinformatics tools and application of advanced statistics such as artificial intelligence (AI). Machine learning (ML), a subset of AI, is the most promising tool nowadays in search of new clinically relevant patterns and reliable predictive markers of complex diseases [28,37]. The underlying genetic predisposition to IBD has not been completely revealed employing only the candidate gene approach or genome wide association studies (GWAS). For that reason, there is great interest in the application of AI in IBD research with the goal to improve: patient identification, differential diagnosis, disease risk prediction and clinical outcomes and classification of disease subtypes, as well as identification of disease biomarkers that could be targeted for advancing therapeutic management (Figure 1) [1,28,38,39].

Besides omics, different clinical measurements used for IBD diagnosis and tracking of the disease status, such as fecal calprotectin, blood parameters, serum C-reactive protein, endoscopic and/or medical imaging, possess a large potential that could be exploited in machine learning modeling. A number of studies analyzed usage of clinically valuable traits in IBD diagnostic, prognostic and therapeutic outcome predictions [40,41,42,43,44,45]. For instance, it has been demonstrated that machine learning algorithms employing laboratory and age data outperformed drug metabolic measurements in predicting the response of IBD patients to thiopurines [41]. A promising machine learning utility in IBD is expected for artificial-intelligence-assisted medical images analysis, which is a more objective and computable technique that can automate and improve intrinsically subjective endoscopic evaluations [46,47]. This could help less experienced endoscopists and reduce interobserver variability. Essentially, the real potential of personalized medicine lies in integration of clinical and multiomics data.

This review discusses machine learning application in IBD with the focus on studies that explored genomic and transcriptomic data. First, the basic concepts of machine learning and the foundation of the most used algorithms were explained. Then, we evaluated the representative studies that employed machine learning on genomic and transcriptomic datasets for predicting IBD clinical outcomes or identification of novel risk genes. Finally, we argued the future perspectives of AI in IBD research and prerequisites for its successful translation into clinical practice.

## 2. Machine Learning Approaches

Machine learning is an important area of AI that provides a machine with an ability to learn from experience or find patterns in the data without being explicitly programmed. ML employs self-learning algorithms (set of rules) to solve classification and regression problems (supervised learning) or to find hidden patterns (unsupervised learning) in data. Short descriptions of the common terms related to machine learning used in this review are summarized in Table 1.

Supervised machine learning algorithms are used to solve classification problems or predict response value (regression) based on historical, example data. Example data (the training set) contains labeled instances (usually human subjects in healthcare applications), which means that both input (features or explanatory variables) and output, which is the phenotype of interest (the response variable), are known. Using ML, researchers can obtain an approximate function or a model that successfully differentiates between classes or predicts the numerical value of the response variable. In the designed model, the underlying codependency between the input and output variables often is mathematically complex and not easily interpreted. Successful models should give accurate predictions when applied to novel instances which have not been used for model training. In practice, different types of algorithms are used to learn the model function [48]. The most commonly used classification and/or regression algorithms in IBD research are linear algorithms, support vector machines (SVM), k-nearest neighbors, decision trees, Bayesian algorithms and artificial neural networks [37,49,50,51] (Figure 2).

Unsupervised learning deals with unlabeled data, and the aim is to group instances according to similarity and to find structures within the data. This approach is particularly useful when multiple input variables are included because researchers are unable to visualize and find patterns in hyper-dimensional space. Also, these methodologies are also used for anomaly detection and dimensionality reduction. In life science and IBD research, unsupervised learning algorithms such as hierarchical clustering and the principal component analysis are commonly used [52,53] (Figure 2).

### 2.1. Fitting the Model

In supervised learning, a model represents a function that captures codependency between input variables and the known outputs in order to predict the future unknown outputs of novel instances. Usually, several models are developed in an attempt to obtain a well-fitted model using different algorithms. Algorithms often use iterative protocols to fit a model function to the data. For example, the gradient descent protocol tunes parameters of the model function in small steps toward (local) minimum of the error function, which represents the measure of deviation of the model function from the example data [48]. A model should ideally capture a general trend within the data but not random noise and erroneous measurements. If a model is trained too long (too many iterations) or if it is too complex, the model captures irrelevant details of the example data and does not generalize well when exposed to new data. This leads to overfitting of the model. This problem could be mitigated by limiting the number of iterations or by penalizing the complexity of the model. The commonly used method to assess the (over)fitting of a model is based on splitting the data into training, validation (or development) and testing sets. The training set (exemplary data) is used to fit a model function. The validation set is used to tune the parameters toward a less complex model that would generalize better. Finally, the testing set is used to make the final assessment of predictive performance of the model. Resampling-based techniques such as bootstrap resampling and cross-validation provide an opportunity to use a single dataset for both training and validations. Here, a model is trained of the majority of instances, and only a small part is randomly chosen and set aside for validation. This procedure is usually iteratively performed to obtain robust assessment of the error function. Resampling-based techniques are useful when only a small number of instances is available [54]. Another problem with predictive modeling occurs when neither the training nor validation set is well fitted by a model function, which is called underfitting. This problem is easily noticed by examining predictive performance of the model on the training set. In case of underfitting, other model functions and algorithms should be tried in order to develop a well-fitted model.

Classification model is evaluated according to its accuracy, which is the percentage of correct predictions. Another popular metric used to assess and visualize the predictive performance of classification models is area under the receiver operating characteristic (ROC) curve or AUC. The curve captures codependency between true positive rate and false positive rate of the model. A similar metric, area under the precision-recall curve or AUPRC, captures codependency between precision (or positive predictive value) and recall (or sensitivity). Both AUC and AUPRC provide a single-value metric to evaluate predictive performance of a classification model, which normally ranges between 0.5 (poor predictive performance) and 1 (perfect classifier) [55].

Different algorithms are used to fit a model function to the data. The choice of an algorithm depends on the problem (classification, regression, clustering, etc.) and the dataset (number of features, number of instances, codependency between features, etc.).

### 2.2. Linear Algorithms

Linear algorithms assume linear dependency between input variable(s) and the output. Assumed linear dependency provides a more straightforward interpretation, because the contribution, both the sign and the effect-size, of each input variable to the model is known. Another advantage of linear models is that they are computationally less expensive to develop. On the other hand, linear models might perform poorly if dependency between input variables and the output is not linear, which would lead to underfitting. In the case of IBD research, there is still no strong evidence that nonlinear models outperform linear ones [37,51,56].

Linear regression is one of the most understood machine learning algorithms used to predict continuous output variables. The algorithm develops a linear model function that best fits the data around a straight line (or a hyperplane). Logistic regression, on the other hand, is not a regression instrument; instead, it is used for classification into distinct categories. Here, a linear function is transformed into a sigmoid-shaped line (the logistic function) which best differentiate between categories. Both linear regression and logistic regression suffer from overfitting and challenging interpretation if multiple input variables are included into the model. To deal with these issues, the modified linear algorithms based on regularization, such as the ridge regression (L2), least absolute shrinkage and selection operator (LASSO/L1) and elastic net (both L1 and L2) can be employed (Table 1). These algorithms penalize the complexity of the model function by shrinking coefficients coupled with input variables toward zero. The coefficients of less predictive input variables can shrink exactly to zero (often encountered with L1 regularization), which would effectively exclude those variables from the model. Another regularization strategy is to shrink all coefficients more evenly (L2 regularization), which effectively deals with codependent features [57].

### 2.3. Nonlinear Algorithms

Nonlinear algorithms do not assume the shape of model function and provide more flexibility and, therefore, better opportunity to develop a well-fitted model. However, they are often more computationally expensive to develop, more prone to overfitting and harder to interpret than the linear models. In addition, these algorithms usually require many instances to provide optimal results [48]. Commonly used nonlinear algorithms include decision trees, k-nearest neighbors, support vector machines, naïve Bayes and neural networks (Table 2). Deep learning is a very popular extension of neural network algorithms which employs multiple hidden layers of interconnected artificial neurons stacked between the input and the output [58].

Classification and regression models often suffer from a poor prediction performance either because of overfitting or underfitting the training data. The so-called “ensemble methods” can address these issues by combining predictions from multiple (usually weak) models, which delivers better predictive performance [48]. Among the commonly used ensemble methods are random forest and gradient boosted trees (GBT), both enabling higher prediction performance of decision tree algorithms [37,49] (Table 2).

### 2.4. Clustering Algorithms

Clustering algorithms group instances based on similarities or distance in feature space. As it is an unsupervised ML approach, the number of clusters is not predetermined. Hierarchical clustering iteratively groups instances into larger clusters until being merged into a single cluster. The clustering process is captured in a tree-like structure which expectantly reflects the underlying organization of the data. Bayesian hierarchical clustering is also a bottom-up approach in which statistical testing guides grouping of the clusters [59].

## 3. Machine Learning in IBD Research

ML methods are currently the best tools for dealing with complex omics data in IBD prediction. The main issue in standard association genotype–phenotype studies using omics data is the large number of multiple comparisons that require rigorous statistical methods for avoiding false positive results. Because of that, many potential causal variants with usually small effect sizes are being neglected. By contrast, ML approach is more flexible in recognizing disease patterns regardless of the statistical level of the associated variants [60]. Only a small percent of IBD heritability is currently explained by identified risk loci [12]. The genetic architecture of IBD is polygenic, with both rare and common variants contributing to disease risk. The large effect sizes have single causal variants (*IL10* and *XIAP*), followed by high-risk variants (odds ratio [OR] > 2) (*NOD2* fs1007insC, *CARD9* c.1434+1G>C, *HLA*-*DRB1*, etc.), then medium-risk variants (OR 1.2–2) (*NOD2* Asn289Ser, *IL23R* Val362Ile, etc.), to the common disease susceptibility loci, which have small effect sizes (OR < 1.2), collectively accounting for only a fraction of variance in disease liability [12]. ML has the ability to select predictor genes with small contributions and to capture effects of epistasis (interactions between genes) which is very important for complex diseases. Thus, ML may further resolve genetics of IBD and indicate new relevant pathways, undiscovered before by standard statistics.

There has been an increased interest in recent years in using AI to explore omics data for IBD risk prediction and classification [37,49,51,61]. The designed ML models had variable prediction performance, with AUC ranging from 0.7 to 0.95, depending on the used dataset and applied method (Table 3). The most frequently employed ML methods included penalized regression models, random forest, support vector machines, Bayesian approach and neural networks (Table 3). Even though ML models are often “black boxes”, they could be used for identifying potentially causal molecular patterns of IBD by evaluating the most significant genes/features selected during the process of model training [52,53,62,63,64].

ML algorithms are data hungry and need large sample sizes to obtain the best performance. The whole process of ML assumes iterative steps of model training with validation followed by model testing on the independent dataset. The common practice in the majority of ML studies performed on IBD is to prefilter variants for subsequent modeling [51,65,66]. In this way, high computational costs caused by high-dimensional input data can be reduced while the overfitting problem is avoided. Even so, the best strategy for detecting all causal variants of complex diseases might be the application of sparse penalized models on the whole set of genotyped variants [60].

Taking into account the price of genome and transcriptome analysis, prediction modeling using omics data on large cohorts could be expensive. However, as the price of the NGS and other high-throughput techniques decreases over time, more frequent application of ML using omics data in IBD risk predictions is expected. Still, beside the greater availability of the high-throughput techniques, achieving good predictive results is often limited due to widespread presence of confounding effects, relatively low prevalence of IBD and high heterogeneity of the disease phenotypes [63]. These issues often limit the analyzed sample size or make the dataset less uniform. Large IBD consortiums having collected and analyzed tens of thousands of samples along with promoting open-access data are an extremely valuable source of omics datasets which could be extensively explored in IBD prediction modeling.

### 3.1. Machine Learning Using Genomic Data

Currently, the largest available IBD genomic dataset has been provided by the International IBD Consortium (IIBDC). The IIBDC dataset was used by Chen and colleagues to predict IBD risk scores [65]. This dataset consists of GWAS imputed and Immunochip genotyped SNPs from over 68,000 IBD patients and 29,000 healthy controls that enabled discovery of more than 200 risk IBD loci [17,30]. Immunochip is a custom Illumina assay comprising 196,524 SNPs and small indels selected primarily based on GWAS analysis of 12 autoimmune and inflammatory diseases [30]. In their analysis, Chen et al. varied the methods for estimating IBD risk score, sample size and type of data used for prediction (GWAS or Immunochip). Their study pointed to Bayesian hierarchical clustering as the best performance algorithm. In addition, they showed that Immunochip data had similar prediction performance as GWAS, largely due to the guidance of the initial GWAS for the Immunochip marker selection. This study indicated that the power of genomic CD and UC prediction was mainly due to strongly associated SNPs with considerable effect sizes. Additional SNPs tagged by GWAS arrays and rare variants found on the Immunochip contributed little to prediction accuracy. Other studies as well came to similar conclusions. The inclusion of not only a significant but broader set of variants, as well as the addition of rare alleles in IBD-established genes, did not improve disease risk prediction performance [37,51]. These results were in contrast with the expected potential of ML to reveal genetic variants carrying marginal IBD risk effects.

Wei and coworkers also used the IIBCD dataset [66]. The study yielded an IBD risk prediction model with high performance (AUC 0.860) using the penalized logistic regression method. The authors applied a two-step feature selection strategy: first, features (genetic variants) were filtered after single-association tests by less stringent association significance cutoff (<10^−4^) and taking into account the frequency of the minor allele (>0.01), and then, LASSO (L1) penalization was performed on the remaining variants. Given the size of the dataset, the LASSO penalization approach was chosen because it requires only one parameter to be tuned during the process of optimizations, which decreased the high computational cost of the analysis.

Another study that exploited the IIBDC Immunochip data was conducted by Romagnoni and colleagues [37]. The authors aimed to make predictions of CD probability employing a set of ML methods: penalized logistic regression, gradient boosted trees and artificial neural networks. All ML methods showed AUC values in similar ranges. The slightly increased performance was accomplished using the ensemble method that combines logistic regression, gradient boosted trees and neural network classifiers, indicating that different models can be seen as partially complementary. This study pointed to several important conclusions—that quality control, imputing methods for missing genotypes and coding strategies for input data can affect the performance of the model, inducing artificial increase in the AUC scores [37].

One great example of the community experiment is the critical assessment of genome interpretation (CAGI), which aims to advance ML methods for genotype–phenotype prediction. CAGI provides a platform for assessing training and testing datasets (https://genomeinterpretation.org, accessed on 2 April 2021) which participants can use to make blind predictions. Since 2010, CAGI has presented dozens of datasets, so-called “challenges”. During each challenge, the CAGI organizers release unpublished data and formulate a specific task related to it. After the closure, organizers evaluate performances of submitted predictions, and a conference is organized to discuss results and emerging ideas. This common task framework led to significant insights into ML-related problems.

In the years 2011 (CAGI2), 2013 (CAGI3) and 2016 (CAGI4), researchers tried to distinguish between CD and healthy controls based on whole exome data. CAGI 2, 3 and 4 datasets’ sizes were not large, counting 56, 66 and 111 exomes, respectively. The work on these challenges stressed the critical points of ML application in genomics—discovering hidden biases in datasets, finding the best strategies to reduce data dimensionality and dealing with limited sample size [70,71].

One successful submission in CAGI4 was performed by Pal and coworkers [51]. They reduced the number of predictors by filtering exome data, including only genomic regions previously associated with CD [30,72]. The authors tested four ML algorithms—logistic regression, random forest, naïve Bayes and a neural network—and varied the number of genetic loci incorporated into the model (90 vs. 138). The best performance was achieved with naïve Bayes. A higher number of included loci improved prediction accuracy.

In a recent study by Raimondi et al., the authors designed a novel neural network approach model, CDkoma, to classify CD from healthy controls using CAGI 2, 3 and 4 editions exome data [63]. Initially, the established CD associations were used for selection of predictor genes using PhenoPedia [73]. The authors further dealt with high-dimensionality of the dataset applying efficient encoding strategy. Before entering the model neural nodes, the genetic variants were firstly aggregated at the gene level by counting how many times each type of variant occurs in each gene. This minimized complexity of the training data and the issue of overfitting, making this approach particularly suitable for the small size datasets. Interestingly, this study attempted to “open the neural network black box” and allow a biological interpretation derived from the ML model, even though the neural networks are known to be one of the most difficult ML to interpret.

Similar to the Raimondi study, Wang et al. applied gene-level encoding strategy [64]. For each gene in the set, the gene function score was computed on the basis of predicted functional effects of all its variants. This scoring system was far better compared to the one that calculated the total number of risk variants per gene [63,64]. Wang analyzed the performance of SVM model with leave-one-out cross-validation on CAGI4 as CD-train and CAGI3 as CD-test dataset. Selecting genes in the process of computational feature selection without any previous knowledge of CD biology gave better results than choosing predetermined GWAS genes (AUC 0.75 vs. 0.70, respectively). This suggests that functional effects of variants are more likely to highlight causative signals rather than association signals. Only a few genes appeared both in the feature selection and experimentally derived (GWAS) sets, implying that computational feature selection could identify previously unknown CD-related genes and could be the best choice for analyzing complex diseases where suspect genes are not established or GWAS studies data are not available.

It has been estimated that the prediction of IBD and particularly CD, given its high heritability, should be able to achieve a maximum AUC between 0.96 and 0.98 by genomic profiling (assuming that all risk loci and their effect sizes are known) [74,75,76]. Even though this number seems to be promising, it should be noted that the low prevalence of the IBD limits the utility of genetic prediction. If the prevalence of the disease is low, for instance 1% with theoretic AUC of 0.98, only 12% of individuals who test positive develop the disease [74]. However, IBD risk prediction is hardly ever required for testing in the general population. Subjects who have family history of IBD or are at higher risk of having unresolved gastrointestinal symptoms or undetermined CD or UC diagnosis represent a distinct population in which the incidence of IBD is much higher. Moreover, genetic prediction may be used in existing patients to classify them in disease subphenotypes, to infer course of the disease and treatment response [77]. Therefore, the clinical utility of these models could be more important for higher risk groups and diagnosed patients than for the general population.

### 3.2. Machine Learning Using Transcriptomic Data

Apart from genomics, other fields of omics, such as transcriptomics, have been explored in IBD risk predictions. The search for reliable IBD biomarkers outside of purely genetic studies is emphasized by the fact that all IBD-associated genetic factors identified so far can explain only 20–25% of described cases, a small fraction of IBD variance and variability within subphenotypes [1,78].

Isakov et al. developed ML-based gene prioritization method to differentiate IBD-risk genes from non-IBD genes [49]. The supervised method was generated to produce two outputs—positive, if the gene had previous GWAS established IBD associations, and negative, if the gene had no association with IBD. Each gene was characterized with gene expression data and gene annotation features, which were used to construct the prediction model. Using the selected features, Isakov et al. trained four different ML models to produce gene risk scores: random forest, SVM, gradient boosting and elastic net. The range of each risk score was from 0 to 1, which corresponded with the level of confidence in which a gene is considered to be an IBD risk gene. The method was then used to assign the risk scores to the comprehensive list of 16,390 genes. The model has selected 347 genes with high prediction scores for IBD risk; among which, 163 were already known IBD genes, 117 genes had at least one publication associated with IBD, and for the residual 67, no existing research was found. This is a good example of how vast data existing in the public domain may be used to discover novel IBD-associated genes.

A recent study by Smith et al. used comprehensive transcriptomic data from the recount2 [79] database to address different predictive problems related to phenotype classification [56]. The recount2 database contains the analysis-ready RNAseq count data from genotype tissue expression (GTEx) project, the cancer genome atlas (TCGA) and the sequence read archive (SRA). The aim of the Smith study was to test ML in predicting numerous binary and multiclass phenotype outcomes; among which, two were related to IBD. Particularly, they used colon tissue transcriptomic data to classify three types of CD-B1 (inflammatory), B2 (stricturing) or B3 (penetrating/fistulating) behavior as well as to predict etrolizumab response in UC patients. The study analyzed the impact of normalization techniques, different sizes gene sets and ML techniques such as logistic regression, random forest and k-nearest neighbors. It was demonstrated that multivariate predictors outperformed predictors based on the single gene and that larger gene sets were more informative compared to smaller ones. In addition, L2-regularized regression applied to the centered log-ratio transform of transcript abundances was shown to be the best choice for predictive analyses using transcriptomic data.

Unsupervised ML methods could be utilized to categorize patients without any previous assumptions and obtain potently better classifications than existing ones. For instance, hierarchical clustering has been used to assess the classification of operated TNF-naïve CD patients using transcriptome signature in ileum mucosa [52]. It has been shown that patients with a Rutgeerts score of i0 (measure of disease activity at the follow-up colonoscopy) largely segregate together and are independent of patients with scores i1-4. Moreover, i0 vs. i1-4 segregation was better than between i0 and i1 vs. all other scores, even though i0 and i1 are usually considered to be signatures of clinical endoscopic remission. When this differential classification was further analyzed in a random forest model, a set of 30 transcripts was selected as the most influential in the model. Transcripts involved in the regularization of Bcl-2 and Bax-mediated apoptosis, a cell process discussed before as potentially significant for ileal CD subtype categorization [80], were identified among the significant predictors of postoperative remission. The prediction model demonstrated high accuracy: 92–93% of estimates in random forest were correct.

Studies performed on IBD suggest that genetic contribution is weaker in UC compared to CD [81,82]. Thus, using gene expression data for diagnostic, prognostic and classification purposes might be more appropriate for UC than using genomic data. The recent study by Khorasani et al. [53] designed a predictive model to discriminate UC and healthy controls using colonic transcriptome data. Datasets were selected from different studies to reduce the effect of technical conditions, and the training and validation sets were independent. Prior to ML modeling, the authors applied a novel feature selection method in order to reduce input data dimensionality to 32 genes, which were further used in an SVM classifier. The final model was able to classify active and inactive UC from healthy donors with average precision of 1 and 0.62, respectively. From the selected 32 genes, most did not have a direct link with IBD phenotype, and some were related to IBD-associated comorbidities, such as altered blood pressure, cholesterol level or colorectal cancer. One more example where ML modeling with its hypothesis-free approach could be a useful tool to identify novel risk genes whose role in IBD would be investigated further on.

There are many ways to perform the selection of genes that contribute the most to disease classification. Yuan and colleagues searched for the genes expressed in IBD blood samples which could be used to classify CD and UC from non-IBD subjects [62]. They applied a two-step feature selection method. First, the genes were ranked according to their relevance to the sample class label and their mutual redundancy. In the second step, incremental genes/feature selection was used in SVM model with a tenfold cross-validation to obtain most optimal combination of genes for discrimination of UC, CD and healthy samples. This approach yielded a set of 21 genes that could predict a diagnosis with high accuracy (93.7%). The obtained set of genes was extended with an additional 20 genes after evaluating the interaction network of proteins coded by these genes. Gene ontology pathways enriched with identified genes have been recognized before as important to IBD, such as the T-cell receptor signaling pathway, cell activation, and apoptosis.

Independent validation is a critical step in the development of any biomarker, assay or prediction model, in which how well they perform on the unseen data can be tested. The successful validation of IBD biomarkers was demonstrated in Biasci and colleagues’ prospective study [83]. In the previous work of the authors, unsupervised clustering of CD8 T-cells transcriptome data separated IBD patients into two distinct subgroups, which subsequently demonstrated contrasting disease courses. To simplify diagnostic procedure needed for patient stratification, the authors aimed to develop a qPCR test consisting of several of the most significant classifier genes from the whole-blood transcriptome using logistic regression with adaptive elastic net penalty. A list of 39 candidate genes was selected from the top models; however, it was shrunk to 17 genes during the qPCR validation in repeated penalized regression analysis. The 17 genes set (15 informative and 2 reference genes) was further validated in the independent, newly diagnosed group of patients using the qPCR method. The negative predictive value of the established test was very high, which was important for the identification of patients who do not need additional therapy [83]. This is an example of good practices in the development of prediction models that could be easily translated into clinical practice.

Another increasingly explored omics area in IBD is microRNAs [84]. MicroRNAs are more stable than mRNA and easily accessible in blood or urine, which categorize them as promising noninvasive markers for IBD diagnosis. Studies by Hübenthal et al. and Duttagupta et al. used the information on microRNA from peripheral blood to construct prediction models that distinguish between healthy and diseased individuals [67,85]. Hubenthal employed a penalized SVM method for selecting a small set of 16 distinct microRNAs (from a total of 863) which were sufficient for sensitive and specific classification between CD, UC and controls [67]. Duttagupta extracted the signatures of 31 differentially expressed platelet-derived microRNAs that in the SVM model demonstrated high accuracy, specificity and sensitivity in differentiating UC patients from normal individuals [85]. However, the limitations of both studies were small sample sizes and lacked proper independent datasets, which could lead to overfitted models.

Single-cell RNA sequencing technology is increasingly used in IBD research, allowing the detailed analysis of different phenotypes of each cell type. This is very important in IBD research because inflammatory phenotypes of immune cells are enriched in inflamed tissues. In addition, the detection of such cell phenotypes is associated with disease progression and therapy failure, as shown in a study by Martin et al. [69]. In this study, the authors used an unsupervised ML approach—hierarchical clustering—to differentiate between major cell types. The performance of hierarchical clustering using single-cell RNA sequencing data was compared to cytometry analysis, and the results showed a strong correlation between the two methodologies, both in inflamed and uninflamed tissues. Subsequent principal component analysis examined differential cellular subtype frequencies between paired inflamed and uninflamed tissues. The analysis showed that the first two principal components can explain around 73% of variance referred to cellular subtype composition. This approach can help differentiate the cellular composition of inflamed and uninflamed tissue, which could have great potential for clinical application facilitating precise diagnosis, disease localization and therapeutic decisions.

## 4. Future Perspectives

This review has been focused on machine learning applications in IBD research using genomic and transcriptomic data. Beside this, there is a vast potential among metagenomic, proteomic, epigenomic, metabolomic, and even single-cell transcriptomic data that were and would be explored in machine learning modeling [86,87,88,89,90]. The integration of data from individual IBD-relevant ‘omes’ is currently considered as the approach that would significantly improve the understanding of IBD pathogenesis and management [1,13,21]. One of the key challenges in this process is to effectively utilize information obtained in omics studies with patients’ data stored in electronic medical records (biochemistry tests, various imaging data, symptoms at diagnosis and lifestyle specifics) [38,91]. Machine learning approaches offer the ability to effectively deal with the high dimensionality of these data with the final aim to translate discoveries into clinical practice. Clinical biobanks that gather the multiomics data together with clinical characteristics of patients, such as 1000IBD, RISK and PRISM cohorts, are essential for bringing these up-to-date statistical methodologies to their maximum [92,93]. These synchronized collections of patient metadata provide the raw material for a future significant improvement of precise diagnoses, disease monitoring and personalized treatments. However, to reach these objectives, the prospective validation of AI application should be performed in independent IBD cohorts. The benefits of such methodology are relatively easily demonstrated within one study, but it is much harder to replicate these to independent studies due to high genetic and environmental diversity in human populations [68]. In addition, the variation in clinical decisions and therapeutic protocols, as well as complex nature and heterogeneity of IBD, could affect the successful validation of ML in different cohorts. Since the genetic landscape is population specific, it is particularly important to examine feasibility of omics-based AI models among different populations. Meta-analyses of the existing omic studies can aid identification of reliable and replicable IBD classifiers. Selection of the input data in prediction modelling could be directed by the previous genetic insights and results from meta-analysis. Beside this, the standardization of machine learning techniques as well as practicing transparent, open-sourced and easily reproducible computational research improves the development and replication of the machine learning models in biology. Most of all, randomized clinical trials are needed to determine if these prediction models truly improve clinical outcomes, and if they do, cost effectiveness of their usage compared to standard IBD clinical protocols should be assessed. In addition, the ethical issues that follow an individual’s disease predictions should be taken into consideration.

## 5. Conclusions

IBD is a multifactorial, complex and lifelong disease with varying representation in terms of disease type, age of onset, localization, and severity. Accurate diagnosis and prognosis of the disease followed by the right treatment is of the essence for controlling the disease. Emerging technologies provide the means to collect ever more multiomics data from large cohorts of patients. Instead of relying only on a small number of biomarkers, ML algorithms can employ the big data collected from multiomics analyses coupled with electronic health record data to provide more accurate predictions. Large cohorts enable an opportunity to develop more complex ML models able to capture complex dependencies between features resulting in better predictions and detection of novel biomarkers. Still, before being employed in a clinical setting, predictive models should be rigorously tested in independent cohorts and in the settings of clinical trials to ascertain that this approach can indeed bring benefits to IBD patients in terms of prevention, timely and accurate diagnosis and personalized treatment.

## Figures and Tables

**Figure 1 genes-12-01438-f001:**
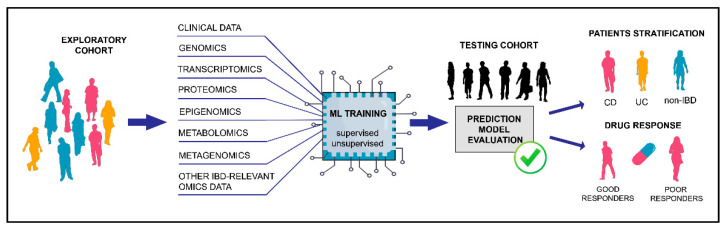
Machine learning using omics data for prediction of clinically relevant IBD outcomes. Omics data from patients with known clinical outcomes (exploratory cohort) can be used as input data in machine learning algorithms during the prediction model training. Performance of the designed model is further assessed on an independent group with unknown outcomes (testing cohort). Machine learning models that have high prediction performance on the testing cohort are well fitted and could be employed for future improved patients’ diagnosis, classification, prognosis and prediction of drug response. ML—machine learning, CD—Crohn’s disease, UC—ulcerative colitis, and IBD—inflammatory bowel disease.

**Figure 2 genes-12-01438-f002:**
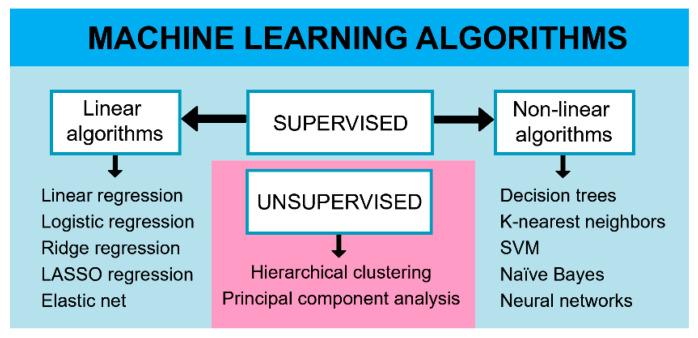
Classification of machine learning algorithms used in IBD research; LASSO—least absolute shrinkage and selection operator; SVM—support vector machines.

**Table 1 genes-12-01438-t001:** Glossary of common terms in machine learning.

Instance	An entity (human subject in healthcare applications) which features are used as inputs for prediction modeling.
Feature	An explanatory variable, such as genetic variant, gene expression, etc. Features are used as input data for prediction.
Machine learning algorithm	Procedure that is run on data to create a machine learning model. It is a set of mathematical optimization functions that minimizes the error of the model function.
Iterations	Machine learning algorithm’s parameters are updated number of times until model reaches desired performance
Classification	Supervised learning technique used to predict a discrete class or category of an instance (disease or healthy subject, good or poor drug responder, etc.).
Regression	Supervised learning technique in which the predicted variable is continuous.
Model fitting	Measure of how well a machine learning model generalizes to data not used for model training.
Penalized regression method	A method used to reduce overfitting of a model. The penalty causes the regression coefficients of less contributive variables to shrink toward zero therefore reducing the number of variables in the model.
Sparse model	A predictive model that includes only the most informative features.
Clustering	Unsupervised learning technique that groups instances by their similarity. The groups are called clusters.
Black box model	Model that is built on complex functions that are not easily interpreted (such as neural networks). Input and output are clear, but the process between is not explainable.
Effect size	A biological measure of the difference or relationship between variables. An OR << 1 or OR >> 1 is indicative of a large effect size.
AUC value	Evaluation metric of a model that ranges from 0.5 (poor classifier) to 1 (perfect classifier).

**Table 2 genes-12-01438-t002:** Classification and regression machine learning algorithms employed in IBD research.

Algorithm	Principle	Usage	Pros and Cons
Logistic regression	Linear model transformed into sigmoid function used as a binary classifier	Classification	Fast to develop; easily interpretable; limited by strong assumptions; prone to overfitting
Linear regression	Classical linear model that employs linear codependency for prediction	Regression (can also be used for classification)	Fast to develop; easily interpretable; limited by strong assumptions; prone to overfitting
Ridge regression	Linear model with L2 regularization	Classification and regression	Linear model with enhanced interpretability and reduced overfitting
LASSO	Linear model with L1 regularization	Classification and regression	Linear model with enhanced interpretability and reduced overfitting
Elastic net	Linear model with both L1 and L2 regularization	Classification and regression	Linear model with enhanced interpretability and reduced overfitting
Decision trees	Prediction based on a tree-like model. Nodes are splitting points of a dataset based on most informative features; leaves are output values.	Classification and regression	Prone to overfitting but can be improved with ensemble methods; interpretable outputs
Random forest	An ensemble method (modified bootstrap aggregation) applied to decision trees. It grows multiple decision trees; output is the average prediction of individual trees.	Classification and regression	High prediction performance; deals with overfitting; requires a large dataset for optimal learning.
Gradient boosted trees (GBT)	An ensemble method (gradient boosting) applied to decision trees	Classification and regression	High prediction performance; hard-to-tune parameters of the algorithm
K nearest neighbors (KNN)	Predicts an output taking into account (k) most similar instances (nearest neighbors)	Classification and regression	Requires a lot of memory to store all the instances; cannot deal with a large number of input variables.
Support vector machines (SVM) classifier	Maximizes margin (decision boundary) between different classes supported by instances that lie near the margin (support vectors)	Classification	Works well with high number of input variables; flexible (allow curved margin by using nonlinear kernels); computationally expensive; limited interpretability
Naïve Bayes	Employs Bayesian posterior probability theorem but assume nondependency between features given the output	Classification	Fast to develop; suitable for large datasets and for making real time predictions; limited by strong assumptions; requires feature selection and transformation
Neural networks	Network of interconnected units resembling the nervous system which renders input information to produce an output.	Classification and regression	High performance; limited interpretability; requires very large dataset; computationally expensive

LASSO—least absolute shrinkage and selection operator.

**Table 3 genes-12-01438-t003:** Studies that explored machine learning for designing IBD prediction models using genomic and transcriptomic data.

First Author and Year [ref]	Machine Learning Algorithm	Predictors/Prediction	Performance	Tested on Independent Cohort	Subjects
Chen 2017 [65]	Bayesian mixture approach	GWAS or Immunochip SNPs data/IBD risk score	CD AUC: 0.75, UC AUC: 0.70	yes	The IIBDGC) cohort—over 68,000 IBD patients and 29,000 healthy controls (4:5 ratio for training and testing, respectively)
Wei 2013 [66]	L1 penalized logistic regression, SVM, gradient boosted trees	Immunochip SNPs data/CD and UC distinction from healthy controls	CD AUC 0.86, UC AUC 0.83	yes	The IIBDGC cohort—~17,000 CD, ~13,000 UC, and ~22,000 controls (randomly divided into 3 folds of equal size for preselection, training and testing, respectively)
Romagnoni 2019 [37]	Logistic regression, gradient boosted trees, neural network and ensemble method	Immunochip SNPs data/probability of CD	AUC 0.8	yes	The IIBDGC cohort—train dataset (34,634 samples), test dataset (17,317 samples)
Pal 2017 [51]	Naïve Bayes	Exome data/CD status	AUC 0.81	yes	Training set: 64 CD and 47 controls (CAGI4); Testing set: 51 CD and 15 controls (CAGI3)
Raimondi 2020 [63]	Neural network	Whole exomes/to distinguish between CD and healthy controls	AUC 0.74–0.83 AUPRC 0.81–0.93	yes	CAGI2, CAGI3, CAGI4 datasets (training and testing)
Wang 2019 [64]	SVM	Whole exomes/to distinguish between CD and healthy controls	AUC 0.7–0.75 AUPRC 0.73–0.80	yes	CAGI4 (training set), CAGI3 (testing set)
Isakov 2017 [49]	Random forest, SVM with polynomial kernel, extreme gradient boosting, elastic net and ensemble method	Data from 2050 genes annotated by the expression (array and RNAseq) and pathway information (categorical terms)/IBD-risk gene prioritization	AUC 0.775–0.829	yes	Intestinal biopsies of 180 CD, 149 UC, 94 colorectal neoplasms, 90 normal tissue (75:25 ratio for training and testing set, respectively)
Cushing 2018 [52]	Unsupervised hierarchical clustering, random forest	Whole transcriptome/identification of markers that could predict postoperative disease activity	92–93% of correct estimates in random forest	no	24 anti-TNFα-naïve patients, 30 anti-TNFα-exposed
Khorasani 2020 [53]	Feature selection algorithm (based on dimension reduction) combined with SVM classifier	Wide expression array data/UC and healthy subjects classification	Active UC AUPRC 1, Inactive UC AUPRC 0.68	yes	Training set: 39 UC samples (active and inactive) and 38 controls; testing set: 97 UC samples (active and inactive) and 22 controls
Yuan 2017 [62]	Feature selection (minimum redundancy maximum relevance and incremental feature selection), SVM-based algorithm (sequential minimal optimization)	Wide expression array data from PBMC samples/CD, UC and normal subject discrimination and candidate gene selection	Accuracy 0.94	no	59 Crohn’s disease, 26 ulcerative colitis, and 42 normal samples
Hubenthal 2015 [67]	Penalized SVM, random forest	miRNAs in whole-blood samples/IBD and control subject distinction	AUC 0.75-1.0	no	40 CD, 36 UC, 38 healthy controls and other inflammation controls (24 chronic obstructive pulmonary disease, 23 multiple sclerosis, 38 pancreatitis and 45 sarcoidosis cases)
Zarringhalam 2014 [68]	Differential expression profile was used to infer upstream regulators using Bayesian approach; posterior probabilities of regulators’ activities were then used in a regularized regression framework to predict outcome	Genome wide expression data/response to infliximab in UC	Accuracy 0.79	yes	Training set: 22 active UC patients (12 responders and 10 nonresponders); Testing set: 24 active UC patients (8 responders and 16 nonresponders)
Li 2020 [50]	Random forest, neural network	RNAseq and microarray expression data/identification of susceptibility genes and establishing predictive model of UC	AUC 0.95; AUPRC 0.97	yes	Training set: 206 UC, 20 normal; Testing set: 53 UC and 21 normal
Martin 2019 [69]	Hierarchical clustering, principal component analysis	Single-cell RNA sequencing data/cell type classification in inflamed and uninflamed tissues	Inflamed tissue (r = 0.96) Uninflamed tissue (r = 0.93) *	no	11 ileal CD patients; samples taken from inflamed and uninflamed tissues

GWAS—genome-wide association study, IBD—inflammatory bowel disease, CD—Crohn’s disease, UC—ulcerative colitis, SVM—support vector machine, AUC—area under the receiver operating curve, AUPRC—area under the precision-recall curve, IIBDGC—The International Inflammatory Bowel Disease Genetics Consortium, *—correlation of cell type frequencies between hieratical clustering analysis applied to RNA profile of a cell and cytometry results referring to that cell.

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
