# Peer review of "Machine Learning Modeling from Omics Data as Prospective Tool for Improvement of Inflammatory Bowel Disease Diagnosis and Clinical Classifications"

_genes, 2021, doi:10.3390/genes12091438_

Round 1

Reviewer 1 Report

This is a comprehensive review of statistical methodology in IBD, specifically focusing on newer techniques such as machine learning. My comments and questions are as follows:

  1. Line 86,87 (references 32,33) - these studies are now quite dated. We need to define more precisely as to what we mean by colonic CD. The question to be answered is - what genes contribute to transmural disease? Please can the authors comment on this.
  2. None of the omics are available in  the clinic. How do we link these to clinical correlates? We need to find these links. Can the authors provide examples from other diseases to provide readers with some real life experience of implementation in clinical practice?
  3. Great care needs to be taken in terms of selecting these cohorts - criteria for mild versus severe, distribution, disease duration - how are these precisely defined and how can the challenge of adequate numbers together with careful selection of cases be achieved? These are some of the reasons as to why replication is so difficult.
  4. The review would benefit by the authors providing clear examples in each sub-section where each technique has either worked well or does not perform well, rather than splitting up methodology and published examples.
  5. Lines 248-258 - please can the authors provide more direct information on effect sizes and heritability taken from the IBD literature. Readers may lose the thread of this section without these details.
  6. Issues in large and expanding datasets are the quality of clinical data and the robustness of the diagnostic criteria. Studying IBD is not like quantitative trait analysis such as blood pressure, LDL - can the authors provide any examples of clinically important quantitative traits or clinical measurements in IBD that may be useful in these analyses?
  7. There is no mention of the transcriptomic work by James Lee (Daniele Biasci, et al. A blood-based prognostic biomarker in IBD. Gut 2019) and colleagues at University of Cambridge with implementation of new tools.

Reviewer 2 Report

The manuscript of Stankovic et al. provides an exhaustive overview of Machine Learning methods and their so far implementation to Inflammatory Bowel Disease (IBD) phenotype prediction based on the available omics data and possible future prospectives.

The text is very well structured, providing an introduction into the IBD diagnosis and prognosis problematics, followed by a comprehensive explanation of existing ML methods and of their application to IBD. It includes several tables and figures, that summarise the existing ML approaches and the main findings, which I find helpful. 

The review is written in a didactic style and might be useful for students and professionals interested in IBD as well as  ML approaches in general and their application to a real case such as IBD.

I consider that the review is ready for publication after some minor English corrections (e.g. in abstract, "and help to address unmet clinical needs" instead of "and help address unmet clinical needs".

Author Response

We appreciate the Reviewer’s comments regarding our manuscript. We have made the suggested correction in the abstract. We also made other grammatical error corrections throughout the manuscript.

Reviewer 3 Report

The review paper on machine learning algorithms in IBD is interesting and nicely describes the machine learning prediction methods. The language is well written and easy to read and understand, however there are some sections that should be better structured and clarified.

Comments:

  1. The introduction could be more concise, and more focused on why the machine learning predictive models are needed in IBD.
  2. Line 158: regression model function is not only used to predict a continuous response variable, for example logistic regression is used for prediction of categorical variables. Please correct this sentence and better clarify the difference between classification and regression usage.
  3. The 2.1. Fitting the model section is rather confusing. The procedure how to design a good practice of using and evaluating machine learning models should be more clearly stated. Tuning the parameters of the model function in small steps towards local minimum is one way how the function selects final parameters. Cross-validation and bootstrapping are resampling procedures used to evaluate machine learning models on a limited data sample and to select the best model. And finally, ROC or AUPROC are usually used as a final result to show the performance of a classification model. All these procedures should be better described and written in more clear and logical way.
  4. Paragraph under the line 315 is redundant, since you have already describe the strength of penalized regression in the 2.1 linear algorithms section.
  5. 2 section might be more concise and focused. Additionally, in the era of Single Cell Sequencing I would recommend to include description on which machine learning methods are being used in Single cell experiments of IBD.

Round 2

Reviewer 3 Report

Thank you for your comments and corrections. I think the manuscript is nicely written and informative. I fully recommend its acceptance.